# Pose Guided Person Image Generation

**Liqian Ma**[1]   **Xu Jia**[2*]   **Qianru Sun**[3*]   **Bernt Schiele**[3]   **Tinne Tuytelaars**[2]   **Luc Van Gool**[1,4]

[1]KU-Leuven/PSI, TRACE (Toyota Res in Europe)   [2]KU-Leuven/PSI, IMEC

[3]Max Planck Institute for Informatics, Saarland Informatics Campus

[4]ETH Zurich

{liqian.ma, xu.jia, tinne.tuytelaars, luc.vangool}@esat.kuleuven.be

{qsun, schiele}@mpi-inf.mpg.de   vangool@vision.ee.ethz.ch

## Abstract

This paper proposes the novel Pose Guided Person Generation Network ($PG^2$) that allows to synthesize person images in arbitrary poses, based on an image of that person and a novel pose. Our generation framework $PG^2$ utilizes the pose information explicitly and consists of two key stages: pose integration and image refinement. In the first stage the condition image and the target pose are fed into a U-Net-like network to generate an initial but coarse image of the person with the target pose. The second stage then refines the initial and blurry result by training a U-Net-like generator in an adversarial way. Extensive experimental results on both $128 \times 64$ re-identification images and $256 \times 256$ fashion photos show that our model generates high-quality person images with convincing details.

## 1   Introduction

Generating realistic-looking images is of great value for many applications such as face editing, movie making and image retrieval based on synthesized images. Consequently, a wide range of methods have been proposed including Variational Autoencoders (VAE) [14], Generative Adversarial Networks (GANs) [6] and Autoregressive models (e.g., PixelRNN [30]). Recently, GAN models have been particularly popular due to their principle ability to generate sharp images through adversarial training. For example in [21, 5, 1], GANs are leveraged to generate faces and natural scene images and several methods are proposed to stabilize the training process and to improve the quality of generation.

From an application perspective, users typically have a particular intention in mind such as changing the background, an object's category, its color or viewpoint. The key idea of our approach is to guide the generation process explicitly by an appropriate representation of that intention to enable direct control over the generation process. More specifically, we propose to generate an image by conditioning it on both a reference image and a specified pose. With a reference image as condition, the model has sufficient information about the appearance of the desired object in advance. The guidance given by the intended pose is both explicit and flexible. So in principle this approach can manipulate any object to an arbitrary pose. In this work, we focus on transferring a person from a given pose to an intended pose. There are many interesting applications derived from this task. For example, in movie making, we can directly manipulate a character's human body to a desired pose or, for human pose estimation, we can generate training data for rare but important poses.

Transferring a person from one pose to another is a challenging task. A few examples can be seen in Figure 1. It is difficult for a complete end-to-end framework to do this because it has to generate both correct poses and detailed appearance simultaneously. Therefore, we adopt a divide-and-conquer strategy, dividing the problem into two stages which focus on learning global human body structure

---

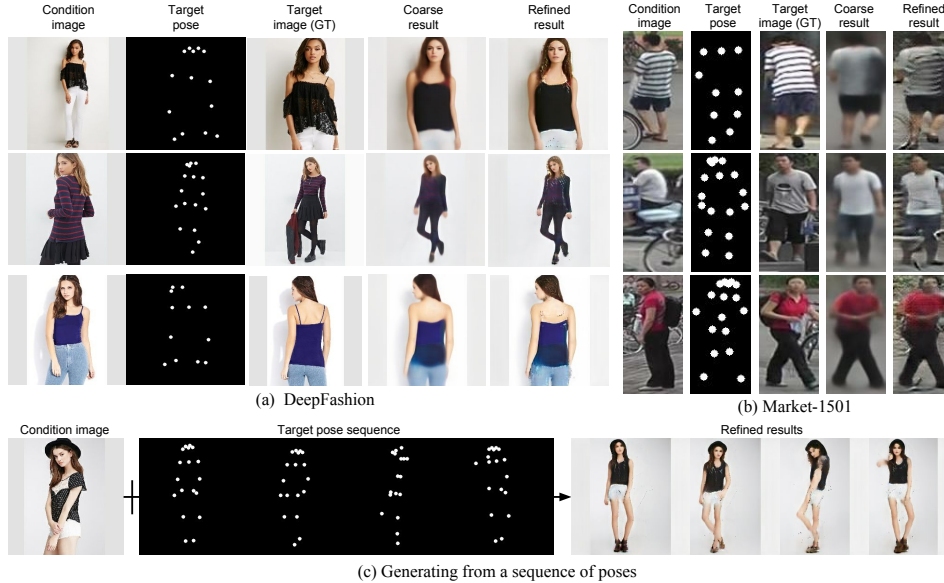

(a) DeepFashion

(b) Market-1501

(c) Generating from a sequence of poses

Figure 1: Generated samples on DeepFashion dataset [16] (a)(c) and Market-1501 dataset [37] (b).

and appearance details respectively similar to [35, 9, 3, 19]. At stage-I, we explore different ways to model pose information. A variant of U-Net is employed to integrate the target pose with the person image. It outputs a coarse generation result that captures the global structure of the human body in the target image. A masked L1 loss is proposed to suppress the influence of background change between condition image and target image. However, it would generate blurry result due to the use of L1. At stage-II, a variant of Deep Convolutional GAN (DCGAN) model is used to further refine the initial generation result. The model learns to fill in more appearance details via adversarial training and generates sharper images. Different from the common use of GANs which directly learns to generate an image from scratch, in this work we train a GAN to generate a difference map between the initial generation result and the target person image. The training converges faster since it is an easier task. Besides, we add a masked L1 loss to regularize the training of the generator such that it will not generate an image with many artifacts. Experiments on two dataset, a low-resolution person re-identification dataset and a high-resolution fashion photo dataset, demonstrate the effectiveness of the proposed method.

Our contribution is three-fold. i) We propose a novel task of conditioning image generation on a reference image and an intended pose, whose purpose is to manipulate a person in an image to an arbitrary pose. ii) Several ways are explored to integrate pose information with a person image. A novel mask loss is proposed to encourage the model to focus on transferring the human body appearance instead of background information. iii) To address the challenging task of pose transfer, we divide the problem into two stages, with the stage-I focusing on global structure of the human body and the stage-II on filling in appearance details based on the first stage result.

## 2    Related works

Recently there have been a lot of works on generative image modeling with deep learning techniques. These works fall into two categories. The first line of works follow an unsupervised setting. One popular method under this setting is variational autoencoders proposed by Kingma and Welling [14] and Rezende *et al*. [25], which apply a re-parameterization trick to maximize the lower bound of the data likelihood. Another branch of methods are autogressive models [28, 30, 29] which compute the product of conditional distributions of pixels in a pixel-by-pixel manner as the joint distribution of pixels in an image. The most popular methods are generative adversarial networks (GAN) [6], which simultaneously learn a generator to generate samples and a discriminator to discriminate generated samples from real ones. Many works show that GANs can generate sharp images because of using

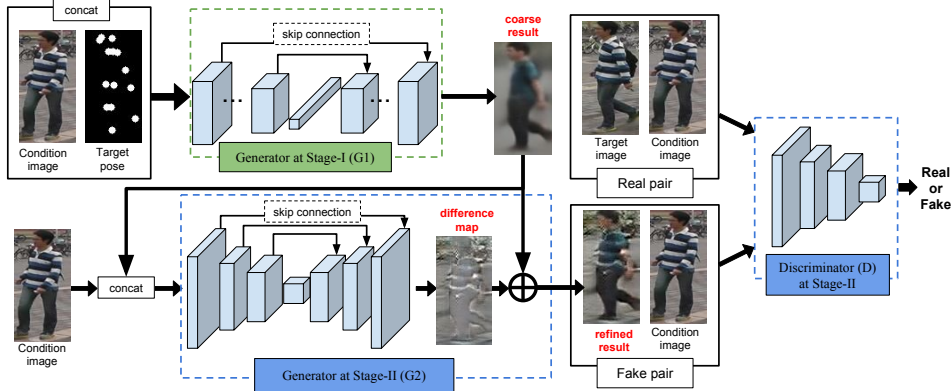

Figure 2: The overall framework of our Pose Guided Person Generation Network (PG²). It contains two stages. Stage-I focuses on pose integration and generates an initial result that captures the global structure of the human. Stage-II focuses on refining the initial result via adversarial training and generates sharper images.

the adversarial loss instead of L1 loss. In this work, we also use the adversarial loss in our framework in order to generate high-frequency details in images.

The second group of works generate images conditioned on either category or attribute labels, texts or images. Yan *et al*. [32] proposed a Conditional Variational Autoencoder (CVAE) to achieve attribute conditioned image generation. Mirza and Osindero [18] proposed to condition both generator and discriminator of GAN on side information to perform category conditioned image generation. Lassner *et al*. [15] generated full-body people in clothing, by conditioning on the fine-grained body part segments. Reed *et al*. proposed to generate bird image conditioned on text descriptions by adding textual information to both generator and discriminator [24] and further explored the use of additional location, keypoints or segmentation information to generate images [22, 23]. With only these visual cues as condition and in contrast to our explicit condition on the intended pose, the control exerted over the image generation process is still abstract. Several works further conditioned image generation not only on labels and texts but also on images. Researchers [34, 33, 11, 8] addressed the task of face image generation conditioned on a reference image and a specific face viewpoint. Chen *et al*. [4] tackled the unseen view inference as a tensor completion problem, and use latent factors to impute the pose in unseen views. Zhao *et al*. [36] explored generating multi-view cloth images from only a single view input, which is most similar to our task. However, a wide range of poses is consistent with any given viewpoint making the conditioning less expressive than in our work. In this work, we make use of pose information in a more explicit and flexible way, that is, using poses in the format of keypoints to model diverse human body appearance. It should be noted that instead of doing expensive pose annotation, we use a state-of-the-art pose estimation approach to obtain the desired human body keypoints.

## 3 Method

Our task is to simultaneously transfer the appearance of a person from a given pose to a desired pose and keep important appearance details of the identity. As it is challenging to implement this as an end-to-end model, we propose a two-stage approach to address this task, with each stage focusing on one aspect. For the first stage we propose and analyze several model variants and for the second stage we use a variant of a conditional DCGAN to fill in more appearance details. The overall framework of the proposed Pose Guided Person Generation Network (PG²) is shown in Figure 2.

### 3.1 Stage-I: Pose integration

At stage-I, we integrate a conditioning person image $I_A$ with a target pose $P_B$ to generate a coarse result $\hat{I}_B$ that captures the global structure of the human body in the target image $I_B$.

**Pose embedding.** To avoid expensive annotation of poses, we apply a state-of-the-art pose estimator [2] to obtain approximate human body poses. The pose estimator generates the coordinates of 18 keypoints. Using those directly as input to our model would require the model to learn to map each keypoint to a position on the human body. Therefore, we encode pose $P_B$ as 18 heatmaps. Each heatmap is filled with 1 in a radius of 4 pixels around the corresponding keypoints and 0 elsewhere (see Figure 3, target pose). We concatenate $I_A$ and $P_B$ as input to our model. In this way, we can directly use convolutional layers to integrate the two kinds of information.

**Generator G1.** As generator at stage I, we adopt a U-Net-like architecture [20], *i.e.*, convolutional autoencoder with skip connections as is shown in Figure 2. Specifically, we first use several stacked convolutional layers to integrate $I_A$ and $P_B$ from small local neighborhoods to larger ones so that appearance information can be integrated and transferred to neighboring body parts. Then, a fully connected layer is used such that information between distant body parts can also exchange information. After that, the decoder is composed of a set of stacked convolutional layers which are symmetric to the encoder to generate an image. The result of the first stage is denoted as $\hat{I}_{B1}$. In the U-Net, skip connections between encoder and decoder help propagate image information directly from input to output. In addition, we find that using residual blocks as basic component improves the generation performance. In particular we propose to simplify the original residual block [7] and have only two consecutive conv-relu inside a residual block.

**Pose mask loss.** To compare the generation $\hat{I}_{B1}$ with the target image $I_B$, we adopt L1 distance as the generation loss of stage-I. However, since we only have a condition image and a target pose as input, it is difficult for the model to generate what the background would look like if the target image has a different background from the condition image. Thus, in order to alleviate the influence of background changes, we add another term that adds a pose mask $M_B$ to the L1 loss such that the human body is given more weight than the background. The formulation of pose mask loss is given in Eq. 1 with $\odot$ denoting the pixels-wise multiplication:

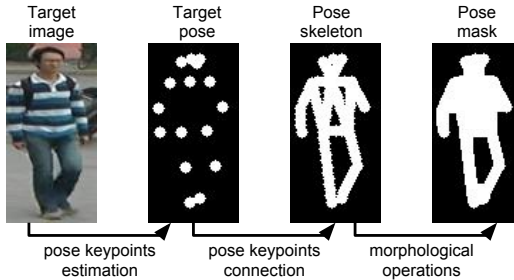

Figure 3: Process of computing the pose mask.

$$\mathcal{L}_{G1} = \|(G1(I_A, P_B) - I_B) \odot (1 + M_B)\|_1. \tag{1}$$

The pose mask $M_B$ is set to 1 for foreground and 0 for background and is computed by connecting human body parts and applying a set of morphological operations such that it is able to approximately cover the whole human body in the target image, see the example in Figure 3.

The output of G1 is blurry because the L1 loss encourages the result to be an average of all possible cases [10]. However, G1 does capture the global structural information specified by the target pose, as shown in Figure 2, as well as other low-frequency information such as the color of clothes. Details of body appearance, i.e. the high-frequency information, will be refined at the second stage through adversarial training.

### 3.2 Stage-II: Image refinement

Since the model at the first stage has already synthesized an image which is coarse but close to the target image in pose and basic color, at the second stage, we would like the model to focus on generating more details by correcting what is wrong or missing in the initial result. We use a variant of conditional DCGAN [21] as our base model and condition it on the stage-I generation result.

**Generator G2.** Considering that the initial result and the target image are already structurally similar, we propose that the generator G2 at the second stage aims to generate an appearance difference map that brings the initial result closer to the target image. The difference map is computed using a U-Net similar to the first stage but with the initial result $\hat{I}_{B1}$ and condition image $I_A$ as input instead. The difference lies in that the fully-connected layer is removed from the U-Net. This helps to preserve more details from the input because a fully-connected layer compresses a lot of information contained in the input. The use of difference maps speeds up the convergence of model training since the model focuses on learning the missing appearance details instead of synthesizing the target image from

scratch. In particular, the training already starts from a reasonable result. The overall architecture of G2 can be seen in Figure 2.

**Discriminator D.** In traditional GANs, the discriminator distinguishes between real groundtruth images and fake generated images (which is generated from random noise). However, in our conditional network, G2 takes the condition image $I_A$ instead of a random noise as input. Therefore, real images are the ones which not only are natural but also satisfy a specific requirement. Otherwise, G2 will be mislead to directly output $I_A$ which is natural by itself instead of refining the coarse result of the first stage $\hat{I}_{B1}$. To address this issue, we pair the G2 output with the condition image to make the discriminator D to recognize the pairs' fakery, $i.e.$, $(\hat{I}_{B2}, I_A)$ vs $(I_B, I_A)$. This is diagrammed in Figure 2. The pairwise input encourages D to learn the distinction between $\hat{I}_{B2}$ and $I_B$ instead of only the distinction between synthesized and natural images.

Another difference from traditional GANs is that noise is not necessary anymore since the generator is conditioned on an image $I_A$, which is similar to [17]. Therefore, we have the following loss function for the discriminator D and the generator G2 respectively,

$$\mathcal{L}_{adv}^{D} = \mathcal{L}_{bce}(D(I_A, I_B), 1) + \mathcal{L}_{bce}(D(I_A, G2(I_A, \hat{I}_{B1})), 0), \tag{2}$$

$$\mathcal{L}_{adv}^{G} = \mathcal{L}_{bce}(D(I_A, G2(I_A, \hat{I}_{B1})), 1), \tag{3}$$

where $\mathcal{L}_{bce}$ denotes binary cross-entropy loss. Previous work [10, 17] shows that mixing the adversarial loss with a loss minimizing Lp distance can regularize the image generation process. Here we use the same masked L1 loss as is used at the first stage such that it pays more attention to the appearance of targeted human body than background,

$$\mathcal{L}_{G2} = \mathcal{L}_{adv}^{G} + \lambda \|(G2(I_A, \hat{I}_{B1}) - I_B) \odot (1 + M_B)\|_1, \tag{4}$$

where $\lambda$ is the weight of L1 loss. It controls how close the generation looks like the target image at low frequencies. When $\lambda$ is small, the adversarial loss dominates the training and it is more likely to generate artifacts; when $\lambda$ is big, the the generator with a basic L1 loss dominates the training, making the whole model generate blurry results[2].

In the training process of our DCGAN, we alternatively optimize discriminator D and generator G2. As shown in the left part of Figure 2, generator G2 takes the first stage result and the condition image as input and aims to refine the image to confuse the discriminator. The discriminator learns to classify the pair of condition image and the generated image as fake while classifying the pair including the target image as real.

### 3.3   Network architecture

We summarize the network architecture of the proposed model PG$^2$. At stage-I, the encoder of G1 consists of $N$ residual blocks and one fully-connected layer , where $N$ depends on the size of input. Each residual block consists of two convolution layers with stride=1 followed by one sub-sampling convolution layer with stride=2 except the last block. At stage-II, the encoder of G2 has a fully convolutional architecture including $N$-2 convolution blocks. Each block consists of two convolution layers with stride=1 and one sub-sampling convolution layer with stride=2. Decoders in both G1 and G2 are symmetric to corresponding encoders. Besides, there are shortcut connections between decoders and encoders, which can be seen in Figure 2. In G1 and G2, no batch normalization or dropout are applied. All convolution layers consist of 3×3 filters and the number of filters are increased linearly with each block. We apply rectified linear unit (ReLU) to each layer except the fully connected layer and the output convolution layer. For the discriminator, we adopt the same network architecture as DCGAN [21] except the size of the input convolution layer due to different image resolutions.

## 4   Experiments

We evaluate the proposed PG$^2$ network on two person datasets (Market-1501 [37] and DeepFashion [16]), which contain person images with diverse poses. We present quantitative and qualitative results for three main aspects of PG$^2$: different pose embeddings; pose mask loss vs. standard L1 loss; and two-stage model vs. one-stage model. We also compare with the most related work [36].

### 4.1 Datasets

The DeepFashion (In-shop Clothes Retrieval Benchmark) dataset [16] consists of 52,712 in-shop clothes images, and 200,000 cross-pose/scale pairs. All images are in high-resolution of 256×256. In the train set, we have 146,680 pairs each of which is composed of two images of the same person but different poses. We randomly select 12,800 pairs from the test set for testing.

We also experiment on a more challenging re-identification dataset Market-1501 [37] containing 32,668 images of 1,501 persons captured from six disjoint surveillance cameras. Persons in this dataset vary in pose, illumination, viewpoint and background, which makes the person generation task more challenging. All images have size 128×64 and are split into train/test sets of 12,936/19,732 following [37]. In the train set, we have 439,420 pairs each of which is composed of two images of the same person but different poses. We randomly select 12,800 pairs from the test set for testing.

**Implementation details** On both datasets, we use the Adam [13] optimizer with $\beta_1 = 0.5$ and $\beta_2 = 0.999$. The initial learning rate is set to $2e$-5. On DeepFashion, we set the number of convolution blocks $N = 6$. Models are trained with a minibatch of size 8 for $30k$ and $20k$ iterations respectively at stage-I and stage-II. On Market-1501, we set the number of convolution blocks $N = 5$. Models are trained with a minibatch of size 16 for $22k$ and $14k$ iterations respectively at stage-I and stage-II. For data augmentation, we do left-right flip for both datasets[3].

### 4.2 Qualitative results

As mentioned above, we investigate three aspects of our proposed PG$^2$ network. Different pose embeddings and losses are compared within stage-I and then we demonstrate the advantage of our two-stage model over a one-stage model.

**Different pose embeddings.** To evaluate our proposed pose embedding method, we implement two alternative methods. For the first, coordinate embedding (CE), we pass the keypoint coordinates through two fully connected layers and concatenate the embedded feature vector with the image embedding vector at the bottleneck fully connected layer. For the second, called heatmap embedding (HME), we feed the 18 keypoint heatmaps to an independent encoder and extract the fully connected layer feature to concatenate with image embedding vector at the bottleneck fully connected layer.

Columns 4, 5 and 6 of Figure 4 show qualitative results of the different pose embedding methods when used in stage-I, that is of G1 with CE (G1-CE-L1), with HME (G1-HME-L1) and our G1 (G1-L1). All three use standard L1 loss. We can see that G1-L1 is able to synthesize reasonable looking images that capture the global structure of a person, such as pose and color. However, the other two embedding methods G1-CE-L1 and G1-HME-L1 are quite blurry and the color is wrong. Moreover, results of G1-CE-L1 all get wrong poses. This can be explained by the additional difficulty to map the keypoint coordinates to appropriate image locations making training more challenging. Our proposed pose embedding using 18 channels of pose heatmaps is able to guide the generation process effectively, leading to correctly generated poses. Interestingly, G1-L1 can even generate reasonable face details like eyes and mouth, as shown by the DeepFashion samples.

**Pose mask loss vs. L1 loss.** Comparing the results of G1 trained with L1 loss (G1-L1) and G1 trained with poseMaskLoss (G1-poseMaskLoss) for the Market-1501 dataset, we find that pose mask loss indeed brings improvement to the performance (columns 6 and 7 in Figure 4). By focusing the image generation on the human body, the synthesized image gets sharper and the color looks nicer. We can see that for person ID 164, the person's upper body generated by G1-L1 is more noisy in color than the one generated by G1-poseMaskLoss. For person ID 23 and 346, the method with pose mask loss generates more clear boundaries for shoulder and head. These comparisons validate that our pose mask loss effectively alleviates the influence of noisy backgrounds and guides the generator to focus on the pose transfer of the human body. The two losses generate similar results for the DeepFashion samples because the background is much simpler.

**Two-stage vs. one-stage.** In addition, we demonstrate the advantage of our two-stage model over a one-stage model. For this we use G1 as generator but train it in an adversarial way to directly generate a new image given a condition image and a target pose as input. This one-stage model is denoted as G1+D and our full model is denoted as G1+G2+D. From Figure 4, we can see that our full model is able to generate photo-realistic results, which contain more details than the one-stage model.

For example, for DeepFashion samples, more details in the face and the clothes are transferred to the generated images. For person ID 245, the shorts on the result of G1+D have lighter color and more blurry boundary than G1+G2+D. For person ID 346, the two-stage model is able to generate both the right color and textures for the clothes, while the one-stage model is only able to generate the right color. On Market-1501 samples, the quality of the images generated by both methods decreases because of the more challenging setting. However, the two-stage model is still able to generate better results than the one-stage method. We can see that for person ID 53, the stripes on the T-shirt are retained by our full model while the one-stage model can only generate a blue blob as clothes. Besides, we can also clearly see the stool in the woman's hands (person ID 23).

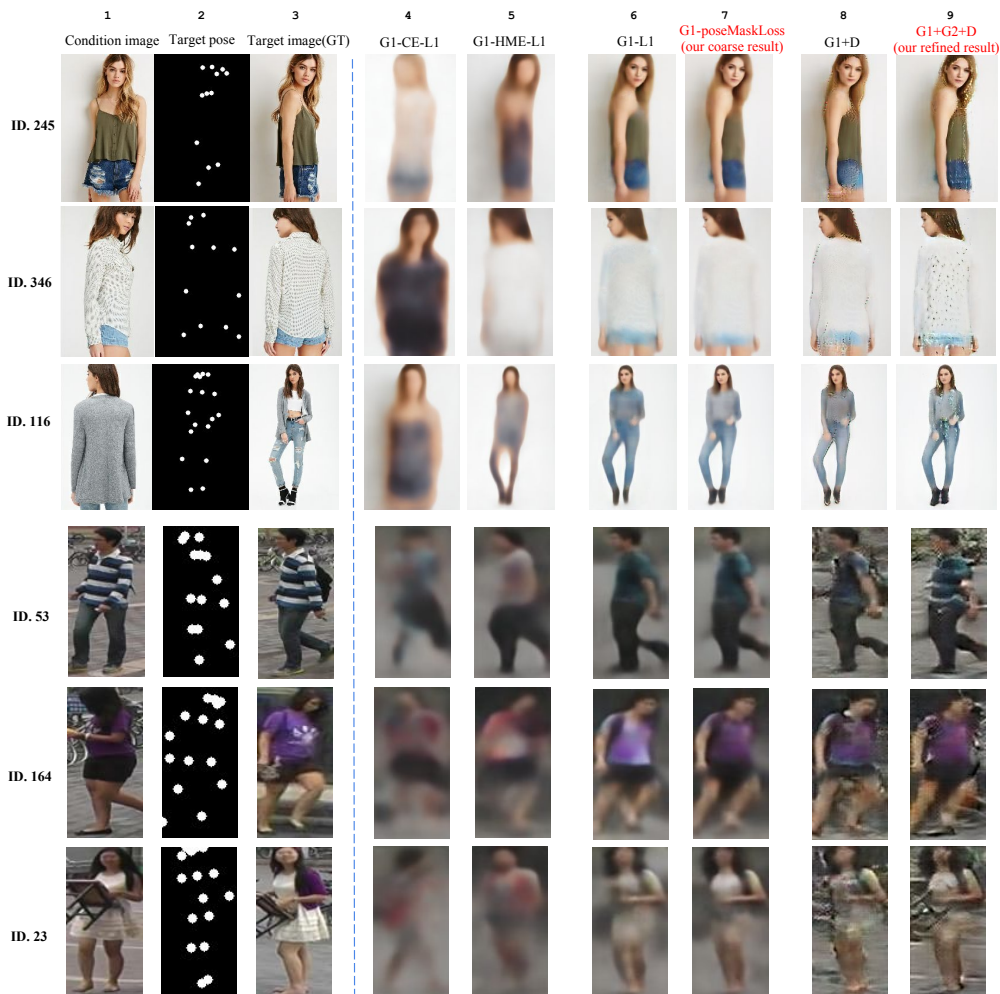

Figure 4: Test results on DeepFashion (upper 3 rows, images are cut for the sake of display) and Market-1501 dataset (lower 3 rows). We test G1 in two aspects: (1) three pose embedding methods, i.e., coordinate embedding (CE), heatmap embedding (HME) and our pose heatmap concatenation in G1-L1, and (2) two losses, i.e., the proposed poseMaskLoss and the standard L1 loss. Column 7, 8 and 9 show the differences among our stage-I (G1), one-stage adversarial model (G1+D) and our two-stage adversarial model (G1+G2+D). Note that all three use poseMaskLoss. The IDs are assigned randomly when splitting the datasets.

## 4.3 Quantitative results

We also give quantitative results on both datasets. Structural Similarity (SSIM) [31] and the Inception Score (IS) [26] are adopted to measure the quality of synthesis. Note that in the Market-1501 dataset, condition images and target images may have different background. Since there is no information in the input about the background in the target image, our method is not able to imagine what the

Table 1: Quantitative evaluation. For all measures, higher is better.

| Model | DeepFashion | | Market-1501 | | | |
| | SSIM | IS | SSIM | IS | mask-SSIM | mask-IS |
|---|---|---|---|---|---|---|
| G1-CE-L1 | 0.694 | 2.395 | 0.219 | 2.568 | 0.771 | 2.455 |
| G1-HME-L1 | 0.735 | 2.427 | 0.294 | 3.171 | 0.802 | 2.508 |
| G1-L1 | 0.735 | 2.427 | 0.304 | 3.006 | 0.809 | 2.455 |
| G1-poseMaskLoss | 0.779 | 2.668 | 0.340 | 3.326 | 0.817 | 2.682 |
| G1+D | 0.761 | 3.091 | 0.283 | 3.490 | 0.803 | 3.310 |
| G1+G2+D | 0.762 | 3.090 | 0.253 | 3.460 | 0.792 | 3.435 |

Table 2: User study results from AMT

| Model | DeepFashion | | Market-1501 | |
| | R2G[4] | G2R[5] | R2G | G2R |
|---|---|---|---|---|
| G1+D | 7.8% | 9.3% | 17.1% | 11.1% |
| G1+G2+D | 9.2% | 14.9% | 11.2% | 5.5% |

new background looks like. To reduce the influence of background in our evaluation, we propose a variant of SSIM, called mask-SSIM. A pose mask is added to both the synthesis and the target image before computing SSIM. In this way we only focus on measuring the synthesis quality of a person's appearance. Similarly, we employ mask-IS to eliminate the effect of background. However, it should be noted that image quality does not always correspond to such image similarity metrics. For example, in Figure 4, our full model generates sharper and more photo-realistic results than G1-poseMaskLoss, but the latter one has a higher SSIM. This is also observed in super-resolution papers [12, 27].

The advantages are also clearly shown in the numerical scores in Table 1. E.g. the proposed pose embedding (G1-L1) consistently outperforms G1-CE-L1 across all measures and both datasets. G1-HME-L1 obtains similar quantitative numbers probably due to the similarity of the two embeddings. Changing the loss from L1 to the proposed poseMaskLoss (G1-poseMaskLoss) consistently improves further across all measures and for both datasets. Adding the discriminator during training either after the first stage (G1+D) or in our full model (G1+G2+D) leads to comparable numbers, even though we have observed clear differences in the qualitative results as discussed above. This is explained by the fact that blurry images often get good SSIM despite being less convincing and photo-realistic [12, 27].

## 4.4 User study

We perform a user study on Amazon Mechanical Turk (AMT) for both datasets. For each one, we show 55 real images and 55 generated images in a random order to 30 users. Following [10, 15], each image is shown for 1 second. The first 10 images are used for practice thus are ignored when computing scores. From the results reported in Table. 2, we can get some observations that (1) On DeepFashion our generated images of G1+D and G1+G2+D manage to confuse users on 9.3% and 14.9% trials respectively (see G2R), showing the advantage of G1+G2+D over G1+D; (2) On Market-1501, the average score of G2R is lower, because the background is much more cluttered than DeepFashion; (3) On Market-1501, G1+G2+D gets a lower score than G1+D, because G1+G2+D transfers more backgrounds from the condition image, which can be figured out in Figure. 4, but in the meantime it brings extra artifacts on backgrounds which lead users to rate 'Fake'; (4) With respect to R2G, we notice that Market-1501 gets clearly high scores (>10%) because human users sometimes get confused when facing low-quality surveillance images.

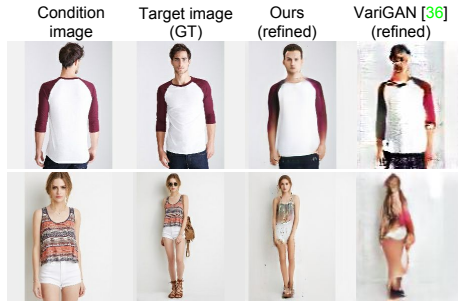
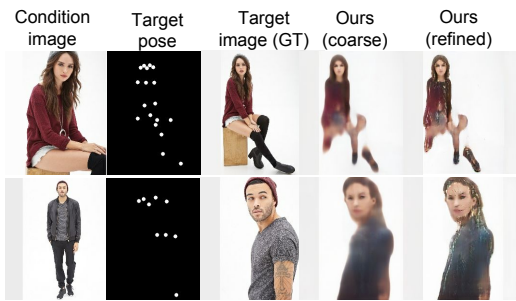

| Condition image | Target image (GT) | Ours (refined) | VariGAN [36] (refined) |

Figure 5: Comparison examples with [36].

Figure 6: Our failure cases on DeepFashion.

## 4.5 Further analysis

Since our task with pose condition is novel, there is no direct comparison work. We only compare with the most related one[6] [36], which did multi-view person image synthesis on the DeepFashion dataset. It is noted that [36] used the condition image and an additional word vector of the target view e.g. "side" as network input. Comparison examples are shown in Figure 5. It is clear that our refined results are much better than those of [36]. Taking the second row as an example, we can generate high-quality whole body images conditioned on an upper body while the whole body synthesis by [36] only has a rough body shape.

Additionally, we give two failure DeepFashion examples by our model in Figure 6. In the top row, only the upper body is generated consistently. The "pieces of legs" is caused by the rare training data for such complicated poses. The bottom row shows inaccurate gender which is caused by the imbalance of training data for male / female. Besides, the condition person wears a long-sleeve jacket of similar color to his inner short-sleeve, making the generated cloth look like a mixture of both.

## 5 Conclusions

In this work, we propose the Pose Guided Person Generation Network (PG$^2$) to address a novel task of synthesizing person images by conditioning it on a reference image and a target pose. A divide-and-conquer strategy is employed to divide the generation process into two stages. Stage-I aims to capture the global structure of a person and generate an initial result. A pose mask loss is further proposed to alleviate the influence of the background on person image synthesis. Stage-II fills in more appearance details via adversarial training to generate sharper images. Extensive experimental results on two person datasets demonstrate that our method is able to generate images that are both photo-realistic and pose-wise correct. In the future work, we plan to generate more controllable and diverse person images conditioning on both pose and attribute.

**Acknowledgments**

We gratefully acknowledge the support of Toyota Motors Europe, FWO Structure from Semantics project, KU Leuven GOA project CAMETRON, and German Research Foundation (DFG CRC 1223). We would like to thank Bo Zhao for his helpful discussions.

## Footnotes

[2]The influence of $\lambda$ on generation quality is analyzed in supplementary materials.

[3]More details about parameters of the network architecture are given in supplementary materials.

[4]R2G means #Real images rated as generated / #Real images

[5]G2R means #Generated images rated as Real / #Generated images

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
