[Supplementary Material · NIPS17_301_PG2_camera_ready_supp.pdf]

# Supplementary materials
# Pose Guided Person Image Generation

Figure 1: Network architecture of generator at stage-I.

Figure 2: Residual block.

## A  Network architecture

In this section, we give the details of our network architecture. Since images in the two datasets have different resolutions, we use slightly different network architecture for them. On the Market-1501 dataset, for generator at stage-I G1, the encoder consists of five residual blocks and one fully-connected layer. The decoder is symmetric to the decoder except for the last layer that generates an image of 3 channels. The network is shown in Figure 1. Each residual block in the network consists of two convolution layers with stride=1 followed by one convolution layer with stride=2 for downsampling except the last block, which is shown in Figure 2.

For generator at stage-II G2, it has similar network architecture to G1 but only three residual blocks in the encoder and decoder, as shown in Figure 3

On the DeepFashion dataset, since the input image has a size of $256\times256$, we use one more residual block than the one used on Market-1501 dataset. That is, we have six residual blocks for encoder and decoder in G1 and four residual blocks for encoder and decoder in G2.

Figure 3: Network architecture of generator at stage-II.

# B    Influence of $\lambda$ at stage-II

We investigate the influence of $\lambda$ to the performance of our two-stage method. As is shown in Figure 4, when $\lambda$ is small, the adversarial loss dominates the training and it is more likely to generate artifacts, e.g., the color of the right arm of person ID. 170 and the faces of person ID. 23 and ID. 53. When $\lambda$ becomes bigger, the generator with a basic L1 loss dominates the training, making the whole model generate blurry results.

Figure 4: Generated results using different values of $\lambda$ at stage-II.

Figure 5: Generated results for one person with various target poses on the DeepFashion dataset.

Figure 6: Generated results for one person with various target poses on the DeepFashion dataset.

Condition Image

Target Poses

Generated refined images

Ground Truth

Figure 7: Generated results for one person with various target poses on the Market-1501 dataset.

## C    Person image generation of arbitrary poses

In this section, we give more generation results conditioned on one person image and a set of different poses. Note that these poses are computed by a pose estimator on random person images in the dataset. From Figure 5 and 6, 7, we can see our model is able to generate photo-realistic results conditioned on multiple poses.

## D    Generated results

In Figure 8 and Figure 9, we show more pose guided person image generation results.

## E    Failure results

In figure 10 and Figure 11, we show some failure cases of our method. The main reasons of failure cases include dramatic scale change, inaccurate pose input, and the rare case of samples.

| Condition Image | Target pose | Target image (GT) | Ours (coarse) | Ours (refined) |
|---|---|---|---|---|

Figure 8: Generated results on the DeepFashion dataset.

Figure 9: Generated results on the Market-1501 dataset.

Figure 10: Failure cases on the DeepFashion dataset.

Figure 11: Failure cases on the Market-1501 dataset.