[Reviews · NeurIPS 2017]

Reviewer 1



Summary: The paper proposes an architecture for generating a images containing a person that combines the appearance of a person in an input image and a specified input pose (using heat maps). Strengths: + the proposed task is novel + novel two stage coarse-to-fine architecture + well written + while the results contain discernible artifacts, they are promising + extensive evaluation of design decisions Weaknesses: - the quantitative evaluation is generally weak; this is understandable (and not a major weakness) given that no suitable evaluation metrics exist Comments: Given the iterative refinement nature of the proposed architecture, it is suggested that some recent work on iterative refinement using cascaded networks be included in the related work, e.g., [a,b]: [a] J. Carreira, P. Agrawal, K. Fragkiadaki, and J. Malik. H man pose estimation with iterative error feedback. In IEEE Conference on Computer Vision and Pattern Recognition (CVPR), 2016. [b] A. Newell, K. Yang, and J. Deng. Stacked hourglass networks for human pose estimation. In European Conference on Computer Vision (ECCV), 2016. - line 110: "we find that using residual blocks as basic component improves the generation performance" In what way? Please elaborate. While the evaluation is generally extensive, I was left curious how using the Stage 2 network with the inputs for Stage 1 and trained adversarially would perform. Is the blurring observed with Stage 1 + GAN due to the intermediate fully connected layer? Overall rating: Based on the novelty of the problem and approach I am confidently recommending that the paper be accepted. Rebuttal: While this reviewer appreciates the attempt at a user study, given its very preliminary nature, I do not find it convincing and thus am discount it in my review rating. The authors are urged to include a substantive human evaluation in their revision. Nonetheless, my positive review of the work still stands.

Reviewer 2



The paper proposes a human image generator conditioned on appearance and human pose. The proposed generation is based on adversarial training architecture where two-step generative networks that produces high resolution image to feed into a discriminator. In the generator part, the first generator produce a coarse image using a U-shape network given appearance and pose map, then the second generator takes the coarse input with the original appearance to predict residual to refine the coarse image. The paper utilizes the DeepFashion dataset for evaluation. The paper proposes a few important ideas. * Task novelty: introducing the idea of conditioning on appearance and pose map for human image generation * Techniques: stacked architecture that predicts difference map rather than direct upsampling, and loss design The paper can improve in terms of the following points to stand out. * Still needs quality improvement * Significance: the paper could be seen one of yet-another GAN architecture * Problem domain: good vision/graphics application, but difficult to generalize to other learning problems The paper is well organized to convey the key aspects of the proposed architecture. Conditioned on appearance and pose information, the proposed generator stacks two networks to adopt a coarse-to-fine strategy. This paper effectively utilize the generation strategy in the dressing problem. The proposed approach looks appropriate to the concerned problem scenario. The difference map generation also looks a small but nice technique in generating higher resolution images. Probably the major complaints to the paper is that the generated results contain visible artifacts and still requires a lot of improvement for application perspective. For example, patterns in ID346 of Fig 4 results in black dots in the final result. Even though the second generator mitigates the blurry image from the first generator, it seems the model is still insufficient to recover high-frequency components in the target appearance. Another possible but not severe concern is that some might say the proposed approach is an application of conditional GANs. Conditioning or stacking of generators for adversarial training have been proposed in the past; e.g., below, though they are arXiv papers. The paper includes application-specific challenges, but this might not appeal to large number of audiences. * Han Zhang, Tao Xu, Hongsheng Li, Shaoting Zhang, Xiaolei Huang, Xiaogang Wang, Dimitris Metaxas, "StackGAN: Text to Photo-realistic Image Synthesis with Stacked Generative Adversarial Networks", arXiv:1612.03242. * Xun Huang, Yixuan Li, Omid Poursaeed, John Hopcroft, Serge Belongie, Stacked Generative Adversarial Networks, arXiv:1612.04357. In overall, the paper successfully proposed a solution to the pose-conditioned image problem, and properly conducts evaluation. The proposed approach sufficiently presents technical novelty. The resulting images still needs quality improvement, but the proposed model at least generate something visually consistent images. My initial rating is accept.

Reviewer 3



This paper develops a GAN-type system to generate a novel-view person images guided by a target skeleton (represented as keypoints). The system contains mostly known techniques such as DCGANs, U-nets, so the contribution is about the specific computer vision application. Pros: * The proposed idea is interesting, the writing is easy to follow, and empirical experiments show that it is able to synthesize images that are realistic-looking at some level. Cons: * As this work is application oriented, it is important to achieve good results. My main complain is ** there is no quantitative evaluation. For example, is it possible to organize user studies to evaluate how many times the generated images can trick human eyes. ** the empirical results are not visually appealing. Take fig.1 for example, the generated images are too small to see. Even this way, there are many noticeable visual errors when comparing the results to the targets. It is thus a bit discouraging to conclude that the results so far are of good quality or close to real images. The same issue stays for almost all the results including the supplementary figures. Personally I feel the authors are at the right direction and the work is promising, while it is yet to stop and declare victory at this stage. After rebuttal: I am still not convinced after reading the authors' feedback. In particular, it has also been pointed by other reviewers that there are still issues regarding quantitative as well as qualitative results, all is important for such a practical oriented work. The rebuttal shows some attempts but they still do not address these concerns to a satisfied degree. I feel it may not be ready to be well presented in a venue like NIPS.